# Tribological Properties Study of Solid Lubrication with TiO_2_ Powder Particles

**DOI:** 10.3390/ma15207145

**Published:** 2022-10-13

**Authors:** Filip Ilie, George Ipate, Florentina Cristina Manaila

**Affiliations:** 1Department of Machine Elements and Tribology, Politehnica University of Bucharest, 060042 Bucharest, Romania; 2Department of Biotechnical Systems Engineering, Politehnica University of Bucharest, 060042 Bucharest, Romania

**Keywords:** CVFC model, solid lubricant, tribological behavior, self-repairing, self-replenishing

## Abstract

Titanium dioxide (TiO_2_), by its tribological behavior, is known as a solid lubricant. TiO_2_ as a solid lubricant, together with tungsten disulfide (WS_2_) and molybdenum disulfide (MoS_2_) decreases friction and excessive wear. By compacting TiO_2_ powder, pellets are formed. Studies and research on the solid lubricant coatings were conducted with success on a tribometer with the possibility of making two simultaneous contacts, pellet/disk, and slider pad/disk. On the disk of a tribometer, we studied the lubrication characteristics of the TiO_2_ powder particles as the third body by intentionally transferring. Results show that the TiO_2_ pellet behaved like an effective oil-free lubricant by self-repairing and self-replenishing. In experiments, a TiO_2_ pellet is intentionally sheared against the surface of the disk, while the slider pad slips loaded on the lubricated surface until the deposited powder film is exhausted. A theoretical model control volume fractional coverage (CVFC) was used to estimate both the wear rate for the lubricated pellet/disk sliding contact and the friction coefficient at the pad/disk separation surface. According to materials properties, disk velocity, pellet and slider pad load, the pellet wear rate, and slider pad friction coefficient, using the CVFC model, can establish the pellet wear rate, and slider pad friction coefficient. The fractional coverage represents a parameter of the CVFC model that varies with time, and it is useful for estimating the film amount from the third body that covers the disk asperities. Model results well enough describe the tribological behavior of the sliding contacts in experiments, both qualitatively and quantitatively. In addition, the theoretical results obtained by modeling and the experimental those obtained in the process of friction, are compared.

## 1. Introduction

The growing necessity of oil-free lubrication generated by advances in friction pair technologies (in the case of the engines) and the reduction of the petroleum oil reserves at the world level assume research for use of other lubricant types, as is the case of the dry lubrication with solid lubricants. In addition, conventional liquid lubricants have proven inadequate for extreme temperature, load, and velocity conditions. Solid lubricants in the form of dry particles (powders) have the advantage of being used in dispersed environments (there, where the liquid lubricants cannot be used), to reduce heat generated by friction.

Powders lubricate the surfaces by coating under load and significant deformation, while growing the speed of accommodation and load-carrying capacity, respectively, reducing friction and wear. Powder lubricants, such as TiO_2_, MoS_2_, and WS_2_, have proven excellent tribological properties [1,2,3,4,5,6,7,8,9]. Several experimental tests have shown that dry lubrication (powder lubricants), can enhance friction pairs lubrication because they decrease the friction and wear below levels met at boundary lubrication [9,10,11,12] of the sliding contacts over very long periods. In the case of solid lubricants (powder lubricants), lubrication is performed by forming transferred films in sprayed, composite, or compact form. The lubricant source used in the work was in compact powder pellet form, intentionally sheared against the rotary disk surface forming a thin transferred film, of the order of the disk surface asperities size. The specific values of mechanical properties presented and in other studies are: particle size of 10–80 nm; particle shape is nanoparticle; specific surface area of 7–11 m^2^/g; a form of nanopowder; density of 3.23–3.78 g/cm^3^; purity, 99.5–99.9%; hiding power ≥100%; matter volatile at 105 °C ≤ 0.50%; residue on sieve (45 μm) ≤0.10%; pH value of aqueous suspension 6.5–8.0; water solution ≤0.5%; oil absorption ≤26.00 g/100 g; resistivity of aqueous extract ≥20 Ω·m; oil dispersion (Hegman) ≥6.00; dry powder whiteness ≥95%.

Solid lubricants (powder lubricants) have some advantages over water-based lubricants, although when mixed with aqueous media they behave not very differently from their solid state, namely: they have the best load resistance and a excellent coefficient of friction and can be used undiluted; provides, lubrication at temperatures (up to 350 °C) higher than water-based and oil-based lubricants; their use is sometimes restricted by their poor resistance to oxidation (they degrade above 350 °C in air, but 1100 °C in reducing environments; they are used as additives in oils or greases, in water and plastics, to confer or increase the properties of their lubrication, but do not reduce the coefficients of wet friction, which are excellent under normal working conditions, but ensure reduced coefficients of friction in extreme conditions, of temperature and/or pressure, or when the wet film no longer exists or no longer forms to demands for various reasons; they can serve as lubricants at solid–liquid interfaces, keeping the surfaces separated and at the same time high fluidity, leading to a very low coefficient of friction even when there is not enough lubricant, reducing heat generation and of the reduced formation of wear particles; their crystals are formed by layers that cleave, slide, easily between them, because the distance between s tracts is higher, the bonding forces being ionic forces, there being no strong forces of attraction between them and it confers the best lubrication properties in the normal atmosphere, thermal stability up to +350 °C–+400 °C (MoS_2_), −270 °C–+650 °C (WS_2_), −300 °C–+800 °C (TiO_2_) in oxidizing environments; the presence of water molecules in the working atmosphere is necessary for these lubrication characteristics, since the absorption of water molecules between the layers of solid lubricants further reduces the forces of attraction between the layers.

In order to have a comparative picture from the point of view of tribological properties of solid lubricants (lubricant powders), the coefficient of friction (being the most suggestive) was considered, whose values are: less than (<) 0.10 for MoS_2_; bigger than (>) 0.03–0.07 for WS_2_; 0.15 for BN and graphite; bigger than (>) 0.04 for PTFE, Teflon, Zonyl, respectively, for TiO_2_ is lower than (<) MoS_2_ but higher than (>) WS_2_, i.e., 0.07–0.09. It is mentioned that these values are for the case of pure lubricating powders. It is mentioned that these values are for the case of pure lubricating powders, but they can vary within greater limits depending on the working conditions (load/pressure), velocity, temperature, friction pair, friction length/road, etc.), especially in extreme conditions.

Therefore, the powder lubricants pelleted represent a novel approach in future applications, for lubricating the friction pairs of the machinery and can be used as a deposition source. Researchers have shown in ref. [13,14,15,16,17,18,19,20,21,22], how pellets can be successfully used for the transferred films on tribo-surfaces. Vazirisereshk et al. studied the mechanism of transferring a thin lubricant film of MoS_2_ compacted, to a rotating disk, both vacuum and in the room environment [23]. Karama et al. in ref. [24] has demonstrated how MoS_2_ powder lubricant pelleted behaved as, of the third body between the high-velocity sliding contacts. The studying of the third body from ref. [24] was extended by Mukras [25] to establish the mass-equilibrium laws and to express wear between the tribo-surfaces, when a third body is formed, using a computerized simulation of wear.

A mass-equilibrium criterion includes adhesive wear law Archard and is applied to establish essential tribological parameters (interfacial friction coefficient pellet/disk and pad/disk, respectively, the wear rate of the pellet).

This paper presents experimental results of the deposition process of a transferred film, from TiO_2_ pellets, simultaneous with that of the lubricant exhaustion process. The CVFC model was developed and applied to quantify the deposition process and to extend the mass-equilibrium concept from ref. [25] and to study the correlation between pellet transferred film (deposited solid lubricant) and the slider pad wear (exhausted solid lubricant) on a tribometer with two pellet/disk and slider pad/disk simultaneous contacts (simultaneous setup).

The focus is on the transferred film of lubricant (third-body particulates (here TiO_2_)) between the sliding contact separation surfaces (pellet/disk and the slider pad/disk). The fractional coverage methodology used in this paper differs from those used by other authors, who applied it to the study of surface coverage by vapor-phase lubrication [26,27,28,29,30]. The theoretical results of the CVFC model are compared and analyzed to the experimental results obtained on the tribometer with two simultaneous contacts (pellet/disk and slider pad/disk).

## 2. Materials and Method

Experimental tests regarding the solid lubrication by self-repairing and self-replenishing were used in the experimental setup (tribometer with two simultaneous contacts) in Figure 1 and entails in-line sliding of a TiO_2_ pellet (compact powder, from which results by shearing, the solid lubricant) and slider pad.

The tribometer is composed of a disk (Figure 1a,b) assembled on a drive shaft by a variable velocity electrical motor, up to a maximum of about 3000 rpm. The disk is executed from titanium carbide (TiC) of 10 mm thickness and an average radius of 50 mm of the test track. The disk of TiC (fixed on the shaft) is driven into rotational movement by the electric motor by a transmission with trapezoidal belts and shaft (see Figure 1a). The sample pellet for testing is maintained in a vertical position by a pellet lubricant holder mounted on the frictionless slide (see Figure 1a). The pellet holder allows the sample pellet to slide freely against the rotating disk and is fixed above on a frictionless slide (low friction slider). The pellet frictional force on the disk is measured with a load sensor attached to the base of the sample pellet holder, with the help of a probe wire. On top of the pellet was placed a linear variable differential transformer (LVDT, see Figure 1a) to record pellet vertical displacement, transformed into the worn material mass, with a resolution of 2 μm.

The pellet/disk and slider pad/disk tribometer was equipped with a load arm to maintain during wear tests the contact between slider pad and disk. The load on the solid lubricating film was taken from the pad loaded with the load (F_s_)_,_ see Figure 1a), by placing some weights at a distance certain from the slider pad mass center. It is important to specify that the sliding pad is executed from the same material (TiC), as the tribometer disk.

The friction and wear parameters (friction force (tangential force) at the slider pad, (F_t_), friction force at the pellet (F_t_’), respectively, the pellet vertical slip (ΔL), see Figure 1a) were recorded with one data acquisition software and displayed on a digital panel.

The TiO_2_ powder (with a particle size of 150–250 nm) was compacted with the help of a compaction system for forming the cylindrical TiO_2_ pellets to a pressure of 34.5 MPa. The pellet sizes were a length of 50 mm and a diameter of 20 mm, and the powder mass of the worn pellets was measured by weighing with the help of an analytical balance with an accuracy of 0.05 g.

## 3. Experimental Procedure

To analyze the powder transferred film mechanism, the wear track in Figure 1b unfolded in the plan, resulting in the schematic setup in Figure 2, where it is observed in-line sliding the TiO_2_ pellet and slider pad. Therefore, pellets fabricated from the compacted TiO_2_ powder are tested at the wear, on the experimental setup (pellet/disk and pad/disk tribometer) in Figure 1a, using the setup of Figure 2 (pellet and slider in-line). The simultaneous results of the pellet/disk and pad/disk tests are compared with those obtained by modeling, using the CVFC model.

This worn material mass vertically was calculated in relation to the wear total mass measured at the end of the experiments based on the mass-equilibrium concept (based on which the CVFC model was developed). As mentioned above, the pellet vertical displacement is recorded with an LVDT (see Figure 1a), which is transformed into the worn material mass, with a resolution of 2 μm.

The measurable sizes from experiments are shown in Figure 1a namely: friction force, F_t_ (tangential force) at the slider pad, friction force at the pellet, F_t_’ and wear by the displacement of the pellet on the vertical, ΔL (see Figure 2a). The load, F_s_ on the slider pad (see Figure 1a) is made by placing weights (at a distance certain from the slider pad mass center) on the load arm from endowing the tribometer with two contacts in line pellet/disk and slider pad/disk. To determine the effective load obtained on the pad, it was necessary to write the equation of the equilibrium moments, since the load, F_s_, did not succeed to be applied in the slider pad mass center.

The parameters of the friction and wear, namely: friction forces, sliding velocity, and pellet vertical slip were recorded at time intervals very short (tenths of a second) with data acquisition software from separation surface pellet/disk and slider pad/disk. Then, the output values of these parameters are displayed on a digital panel (for verification) and with the possibility to the user warn, when between slider pad and disk, the dry friction takes place.

It is observed that the TiO_2_ powder pellet embedded in the lubricant holder and pressed with load, F_p_ is sheared by the rotating disk (see Figure 1a and Figure 2a,b) and by friction, it wears, forming a TiO_2_ particles thin film (third body) on the TiC disk surface. At the same time, the TiC disk is in contact with the slider pad, under the action of load, F_s_, on the lubricated surface (see Figure 1a and Figure 2a). The pellets obtained by compaction had the following dimensions: a length of 50 mm and a diameter of 20 mm.

The TiO_2_ powder identified for research, as a suitable solid lubricant (with average particles size of 150–250 nm) was compacted to the desired pressure (here, at the pressure σ_y_ = 34.5 MPa, see Figure 2a), under the form of cylindrical TiO_2_ pellets, with the diameter of 20 mm and the length of 50 mm. These dimensions together with the pellet mass and density were estimated after compaction, with the help of analytical balance. Then, the pellets (one by one) were introduced into the lubricant holder for testing at friction and wear. The load on the pellet, F_p_ was established at 20 N, and on the slider pad, F_s_ oscillated from 30 N to 200 N (see Figure 2a). During the experimental tests, both the pellet and the slider pad were pressed on the rotating disk with the loads, F_p_ and, F_s_, already established. Additionally, at the same time, the surface of the rotating TiC rigid disk shears the TiO_2_ pellet, because it is pressed with the force F_p_ on the rotating disk, leaving a powder track (deposited by transferring a TiO_2_ powder thin film) on the disk surface, as a result of pellet wear (see Figure 1b).

Thus, the transferred powder film (of TiO_2_) from the pellet on the disk is then distributed by a slider pad on the friction area, until depletion, because its load, F_s_ exceeds the load supported by the powder solid film. Finally, the friction behavior between the separation surfaces pellet/disk and pad/disk of the TiO_2_ powder film, respectively, the delivery rate of the transferred film (the wear rate of the pellet, or the transferred solid lubricant film on the TiC disk surface, being in contact with the slider pad) was analyzed. In addition, with the help of energy dispersive spectroscopy (EDS—Thermo Fisher Scientific, Brno, Czech Republic), we analyzed the transferred solid lubricant film on the TiC disk surface, and the third body is shown in Figure 3

Therefore, the third body obtained by rubbing and shearing the TiO_2_ pellet on the TiC disk is represented by the TiO_2_ powder film (whose TiO_2_ particles are deposited in the valleys between the asperities of the disk, on the pellet/disk contact area (see Figure 2b), until the coating), which acts as a solid lubricant.

EDS analysis is an analytical technique that enables the chemical characterization/elemental analysis of materials of transferred (powder TiO_2_) solid lubricant layer surface and helps to quantitatively measure the coating with TiO_2_ powder.

Scanning the trace left through shearing of the pellet on the disk surface and distributed by the sliding pad during EDS analysis was conducted and the results are given in Figure 3. The spectra of typical EDS (see Figure 3e) show two prominent peaks at around 0.5 eV and 4.5 eV which confirm the presence of titanium (Ti) and oxygen (O), respectively.

From this spectrum, it is observed that TiO_2_ has a stoichiometric ratio of Ti (33%) and O (63.2%). The results of the EDS analyses obtained on the solid lubricant layer surface report that the atomic ratio of Ti/O was close to 1/2 and the two elements of Ti and O were homogeneously dispersed in each TiO_2_ sample.

The experimentation started when the disk reached the established velocity and at the same time, the data acquisition software started. The results are also displayed and can be read on the digital panel, measuring by the displacement on the vertical of the pellet, the friction forces (F_t_ and F_t_’), and, respectively, the average wear depth (ΔL). The wear debris emerging outside the contact surface (from under the separation surface) are collected in a special receptacle. At the end of each race, the disk is prepared (cleaned) for the next race, and the structural integrity of the pellet is visually examined.

Mass of worn material at a pellet for a certain running is established by analytic computing of the removed mass because of pellet length change, using the LVDT system. Additionally, the mass of the worn material of the pellet can be obtained by weighing, with the help of the accuracy analytical balance of 0.05 g, and by calculation with the help of the LVDT system. After the conclusion of the experiments, the mass of worn material calculated through the LVDT system is compared with the one weighted from the analytical balance. The difference between the pellet mass losses computed through the LVDT system compared to that in wear tests was in the range of ±5%.

## 4. Theoretical Considerations

Based on the simplified schematic setup from Figure 2 of the arrangement in line pellet/disk with slider pad/disk, the powder lubricant film of TiO_2_ as a control volume will be analyzed. From the simplified scheme (Figure 2), it is observed how the TiC disk in contact with the TiO_2_ pellet under the action of load, F_p_, moves with sliding linear velocity v, and by friction and shear form a solid lubricant film from powder TiO_2_ (see Figure 2a). At the same time, the solid lubricant film from powder TiO_2_ (see Figure 2a) deposited on the TiC disk surface is in contact and the slider pad under the action of load, F_s_. In addition, Figure 2b shows the pellet simplified schematic setup, where the disk surface can be seen, whose asperities are considered excessively high and shears the TiO_2_ pellet.

The TiO_2_ particulates as the third body obtained by the shear of the pellet have deposited in the valleys between the disc’s asperities the contact route (contact trace), until coverage (i.e., the transferred film as the third body). In principle, to elaborate, a tribological model of the dry lubrication process was included in the control volume, around the transferred film as the third body. Then, we adopted the wear and third body concept in accordance with ref. [9]. While the pellet is worn (see Figure 2b), the TiO_2_ powder particles deposit until cover disk surface asperities with a solid lubricant thin film (TiO_2_ powder), and the control volume grow up from the lowest valley to the highest asperities.

To surprise the tribological behavior, qualitatively and quantitatively, and to quantify the amount of film in the third body, it was necessary to cover the disk asperities on the contact area (see Figure 1b and Figure 2b), to be able to apply the CVFC model. To implement the CVFC model, the following assumptions are necessary:

(1)The pellet/disk separation surface topography is played by the nominal pellet flat surface, which is in contact with the rough surface of a disk with the composite asperities.(2)The disk surface topography little relatively varies in ratio to the asperities maximum height, h_max_ (see Figure 2b);(3)The film transfer process exists at the level of the asperity scale;(4)The friction response of the slider/disk and pellet/disk separation surfaces is mainly a function of the fractional control volume of transferred film covering the disk surface;(5)The disk surface asperities on the track route of powder film, scanned by the slider pad, are filled up concomitantly as the whole track area is lubricated.

Other researchers have developed and applied other relation forms for fractional coverage in the modeling of the lubrication processes [25,26,27,28,29]. The model (CVFC) used in this paper, considers and admits the smooth slider pad, and the friction response is mainly dependent on the thickness (height) variation, h of the transferred solid lubricant film, that covers the disk asperities (see Figure 2b). Thus, the schematic setup in Figure 2b considers the control volume which surrounds the asperity bodies and valleys, together with the transferred particles of the third body from the pellet and indicated by the dotted lines. The supply source of the film of the third body is continuous and comes from the pellet. However, there are two situations when this is depleted: (1) at the front (leading edge) edge of the pellet and (2) at distance from the pellet, i.e., at the slider pad.

Considering the law of mass conservation, it can be written:SR = IR − OR(1)
where SR is storage rate, IR—input rate, and OR—output rate.

With the help of Archard’s law of adhesive wear, in the form:(2)$$\frac{dV}{dt}=K\cdot F\cdot v$$
the inlet (IR) or outlet (OR) rates of the powder lubricant in Equation (1) can be interpreted, where dV—the volume wear rate derivative, V in relation to the time variation, dt of the sliding time, t, F—normal load applied, v—sliding velocity, and K—dimensional wear rate (wear intensity between two different materials (here TiO_2_ on TiC), and in some situations could be only between the same type of material (in the case of the TiO_2_ pellet on the TiO_2_ third body)).

Applying Archard’s adhesive wear law from Equation (2) and the definition of fractional coverage, x = h/h_max_ to the mass conservation law in Equation (1), becomes:(3)$$A\frac{dh}{dt}=K_{p}\cdot F_{p}\cdot v\left(1-\frac{h}{h_{max}}\right)-K_{p}^{\prime }\cdot F_{p}\cdot v\cdot \frac{h}{h_{max}}-K_{s}^{\prime }\cdot F_{s}\cdot v\cdot \frac{h}{h_{max}},$$
where A—the cross-sectional area, dh—the derivative of height (thickness), h of the film of the third body in relation to the time variation, dt, h—the local thickness (height) of the film of the third body, h_max_—maximum asperities height of TiC disk, on the contact track route (see Figure 2b), K_p_—the wear rate of the pellet/disk separation surface, while K’_p_—the wear rate of the third body, due to shearing of the pellet, respectively, K’_s_—the wear rate slider pad.

Because the fractional coverage x = h/h_max_ is a dimensionless size that represents the lubricant fraction (the third body particles) that covers the disk surface asperities, and by replacement in Equation (3), it is obtained:
(4)$$h_{max}A\frac{dx}{dt}=K_{p}\cdot F_{p}\cdot v\left(1-x\right)-K_{p}^{\prime }\cdot F_{p}\cdot v\cdot x-K_{s}^{\prime }\cdot F_{s}\cdot v\cdot x,$$
where dx—the derivative of the fractional coverage, x in relation to the time variation, dt, x = h/h_max_—the fractional coverage of the third-body film.

When the disk asperities are completely covered, then the transferred film height (thickness) is h = h_max_, resulting that x = 1. In the case when no coverage exist with lubricant, then h = 0, result that, x = 0 and represents the initial condition (x(0) = 0).

Thus, the CVFC mathematics model is define of Equation (4) together with x(0) = 0. By solving Equation (4), the CVFC model allows obtaining the fractional coverage, *x* depending on the time, t, i.e., x(t). If the time t is constant, then obtain the steady-state fractional coverage (when the wear rate becomes constant relatively).

Researchers Hershberger, Hichri, Ye, Yoo, and Wornyoh, et al. [28,29,30,31,32], for friction coefficient determination and to obtain information about this, have used other forms of fractional coverage. In this paper, adopting after ref. [28], linear mixtures rule can be defined by the pellet, μ_p_ and slider pad, μ_s_ friction coefficients, thus:(5)$$\mu_{p}=x\cdot \mu_{lub,p}+\left(1-x\right)\mu_{dry,p}$$
(6)$$\mu_{s}=x\cdot \mu_{lub,s}+\left(1-x\right)\mu_{dry,s}$$

These friction coefficients (μ_p_, μ_s_) for lubricated conditions become μ_lub,p_ and μ_lub,s_, while for un-lubricated conditions (dry) become μ_dry,p_, and μ_dry,s_. To obtain the coefficient of the pellet in steady-state wear, Equation (4) is re-written only for the pellet (alone), where the multiplication $h_{max}\cdot A\cdot \frac{dx}{dt}=\frac{dV_{p}}{dt}$, becomes:(7)$$\frac{dV_{p}}{dt}=K_{p}F_{p}v\;(1-x\;(t)).$$

Thus,
(8)$$Vp_{,}^{\mathrm{total}}=\int_{0}^{t_{s}}K_{p}F_{p}v\left(1-x(t)\right)dt,$$
where dV_p_—the derivative of the pellet wear volume, V_p_ in relation to the time variation, dt, and V_p_,_total_—the total pellet wear volume in the total sliding time, t_s_.

Replacing Equation (4) with the result from Equation (8) the wear rate at the steady-state, K_p_, is obtained namely:K_p_ [m^3^/m·kg] = V_p,total_/(F_p_/g)t_s_v,(9)
with g—gravitational acceleration. In addition, the result from Equation (8) allows us to determine the wear rate and at any time, t_s_ = t.

## 5. Results and Discussion

The results obtained from the pellet and slider pad on disk in-line tests are in concordance qualitatively and quantitatively with the CVFC model. During the experiments, a pellet of TiO_2_ was intentionally sheared against the TiC disk to deposit and realize a solid lubricant film (powder of TiO_2_), as a lubricating track, delimited by the compact powder pellet in line with the slider pad that slip over the solid lubricant film deposited by the pellet (see Figure 1b and Figure 2). The friction and wear at the velocities 3.1–31.4 m/s show the representative limits for other experiments of type pellet/disk with slider pad/disk in-line (see Figure 2). The results obtained describe the tribological behavior of the pair pellet and slider pad/disk, in the same conditions: powder compaction tension, σ_y_, contact pressure, p_c_, sliding velocity, v, load on pellet, F_p_, and load on slider pad, F_s_. To assess the behavior of the friction and wear of the powder solid lubricant film between the two sliding contacts, the friction coefficients were also measured at the separation surfaces of pellet/disk, μ_p_, and slider pad/disk, μ_s_ (via the friction forces). Additionally, we measured the vertical pellet wear that transformed into wear mass loss.

Figure 4 shows graphically the experimental results for the friction coefficients (μ_p_ and μ_s_, Figure 4a), respectively, pellet wear (Figure 4b) with the sliding distance (friction length).

The experimental results were obtained in the same conditions of tension (σ_y_ = 34.5 MPa), contact pressure (p_c_ = 0.15 MPa), and sliding velocity (v = 7.85 m/s). Although numerous tests have been performed on tribometer, the authors have decided to present the friction and wear results for a test situation in Figure 4.

It is noticeable that both the friction coefficients and pellet wear have a normal evolution with the sliding distance (friction length). The variation of the friction coefficient, μ_p_ vs. sliding distance (friction length) is not evident in the tribological behavior of the slider pad. However, a closer examination of μ_p_ can be useful in distinguishing the tangential forces indispensable to detaching the TiO_2_ powder particles of pellet [31,32,33,34,35], which leads to its wear. All these statements prove the need to compare the theoretical friction coefficients with the experimental, respectively, and with the theoretical wear rates with those of the experimental.

Therefore, the experiment results from Figure 4a show a high friction coefficient, μ_p_ ≈ 0.28, at the pellet/disk separation surface, when its wear process is initiated. The friction coefficient, μ_p_ gradually falls up to a minimum value of ~0.255 corresponding to the steady state, with small oscillations, as the pellet is worn to form the solid lubricant film. In addition, Figure 4a shows that the friction coefficient, μ_s_ at the separation surface of the slider pad/disk begins from a high value of ~0.26 (but smaller than, μ_p_, when the friction is dry), after which decreases until it reaches the value of ~0.11, at the steady state. During disk rotation, the slider pad friction coefficient, μ_s_ begins to decrease until it touches the lowest value during the experiment, μ_s_ ≈ 0.08, and the sliding distance traveled is relatively short (~0.75 km).

Then, a slow increase takes place, μ_s_ until 0.12–0.16 and the sliding distance of 1.6 km (steady-state, because the running stage has not been finished yet, and then its oscillations have a greater amplitude), and the wear continues to grow on this distance. The evolution of the friction coefficients and the wear take place in two stages:

(1)Run-in, the friction coefficient decreases until it touches a minimum value at the distance of ~0.7 km, as wear increases (so, in opposite directions), then grows rapidly, both the friction coefficient and wear (that continues to grow both in the same direction) until to a sliding distance of ~1.6 km(2)Steady-state (after the distance of 1.6 km), when the friction coefficient and wear evolve in opposite directions (friction coefficient decreases easily, and the wear increases relatively easily with the distance) when there is tendency to stabilize theirs.

Then, the friction coefficient, μ_s_ begins to decrease relatively easily with oscillations relatively small and stabilizes at the value of 0.11–0.13 until the sliding distance of 7 km (see Figure 4a). At the same time, wear increases rapidly, until a sliding distance of 1.6 km, after that it increased relatively slowly, until a sliding distance of 7 km (see Figure 4b). During the run-in period, the rapid increase in wear can start right from negative values, which proves that there is a transfer of material. The steady-state condition was reached at approximately 1.6 km, for the considered test, at which μ_s_ ≈ 0.14, which shows that the slider pad, after the steady state and during the 5.4 km did not influence the experiment (see Figure 4a). The mass loss by pellet wear depending on sliding distance (friction length) is presented in Figure 4b. The corresponding wear as the mass loss of pellet at steady-state is approximately 0.07 g, then it had slow growth, approximately linear for the next 5.4 km (after steady-state), i.e., existed a continuous supply of solid lubricant film (powder), to lubricate the slider pad/disk contact. In other words, the powder solid lubricant supply supported the slider pad load, and the loss mass by wear increased sharply (during the transient period) until steady state (~1.6 km), with sliding distance, after that the wear rate increased relatively linearly on the rest of the sliding distance (Figure 4b). Referring to the friction coefficient, μ_s_ it is observed that between sliding distances 0.9 to 1.6 km increased (that is a normal process until at steady state), and between 1.6 to 7 km the μ_s_ decreased slightly (see Figure 4a). Interestingly, note that in the same intervals of the sliding distance, the corresponding wear increased fast until the steady state, then continued to increase slightly (see Figure 4b), which proves inconsistent at steady-state wear rate (responsible may be anisotropy [19]). On the other hand, the pellet wear rate increased linearly, relatively slightly on the sliding distance from 2 to 7 km, which shows us that there was a continuous powder lubricant and adequate supply and was depleted until sliding contact between the slider pad/disk, leading to a lower and relatively constant friction coefficient, of it.

It is obvious that the lubrication degree from the slider pad/disk separation surface and implicit the friction coefficient, μ_s_ is controlled by the supply amount with TiO_2_ powder particle from the pellet. The need for lubricant supply probably suggests increasing the pad load and thus the friction coefficient also grew. Hence, as the solid lubricant film is depleted, the pellet wears rate increases to compensate for the lack of lubricant film present on the disk.

In this way, the lubrication process assumes to replenish or repair the transferred film as required and thus demonstrates how the solid lubricant particles (powder) as the third body behave tribologically [36,37,38]. A comparison between theory and experiment is shown in Figure 5, and, respectively, the correlation between these.

Figure 5a illustrates that the friction coefficient between slider pad/disk both by the modeling (theory) and experiment increases as slider pad load increases, before leveling. Finally, Figure 5b demonstrates the same situation (at the slider pad/disk separation surface both by the modeling (theory) and experiment), but for the theoretical and experimental wear rate (Kp) of the pellet (i.e., increases with the slider pad load).

Therefore, it is demonstrated that from the friction, the pellet/disk self-replenishes and self-repairs the depleted solid lubricating powder film, and from the compared results, the following is noticed:

-Theoretically, both the friction coefficient variation, μ_s_ and wear rate variation, K_p_ variation is a curve, while the experimental variation is linear (stepped for the friction coefficient and continuously for the wear rate) depending on slider pad load;-Theoretically, the friction coefficient values, μ_s_ are higher than those experimentally, while for the wear rate, K_p_*,* these are the other way around.

At the same time, the tribo-system with two simultaneous processes indicates a suitable source for the continuous supply with the solid lubricant of the sliding contacts.

The theoretical and experimental results explain once more that the solid lubrication with TiO_2_ powder particles through self-replenishing and self-repairing definitely contributes to the durability growth of the friction pairs.

The response to friction and wear of the powder solid lubricants is also influenced by the environmental conditions (relative humidity and temperature), which could certainly explain the differences between theoretical and experimental results. Thus, by including these variables in the model CVFC, it could be improved. Additionally, the differences between theory and experiments at higher loads could be explained when the temperature by friction increases and the relative humidity is changing.

In addition, in support of the above are the images of the surfaces of the TiO_2_ pellet, the disk on the contact area, and the sliding plate before and after testing in the form of photographs/micrographs in Figure 6.

It is observed what the contact surfaces look like in the friction–wear process before testing (Figure 6a–c) and how the same surfaces look after testing (Figure 6d,f,g). It is mentioned that the AFM image (see Figure 6b) of an area on the path (lubrication track) of the TiO_2_ powder film as a solid lubricant was made in order to highlight the condition of the TiC disk surface (the existence of the surface asperities), and after testing she will show that in Figure 6f, which proves that the process of friction, shearing, and wear of the TiO_2_ pellet and of friction of the sliding pad realizes the refilling and repair of the spaces between asperities.

In Figure 6d it can be seen the existence of several scratches and deeper mini-furrows on the front surface of the TiO_2_ pellet resulting from the friction–wear process (as proof that it was worn) compared to the image from Figure 6a. When the TiO_2_ powder is a lubricant, the used surface of the pellet shows whitish-grey sediments, which highlights the presence of Ti together with O and C (the main components of TiO_2_ and TiC), resulting from shearing of the pellet on the TiC disc, as seen in the EDS from Figure 6e. The surface of the TiO_2_ powder film deposited on the lubrication track of the TiC disk (see Figure 6f) shows an agglomeration of TiO_2_ particles, with no obvious signs of wear, because it is continuous, and the disk asperities were completely covered through self-repairing and self-replenishing. Instead, the surface of the slider pad from Figure 6g shows a glossy surface, because it worked on the TiO_2_ powder film, as a solid lubricant and it was polished (which it distributes on the lubrication track (contact area) performing at the same time and the self-repairing and self-replenishing, after which the film was depleted).

## 6. Conclusions

The tests regarding the TiO_2_ powder lubricating were carried out to experiment with the possibility of developing a self-replenishing and self-repairing lubrication process.

The results in a tribo-system with simultaneous processes indicate a suitable source for the continuous supply with the solid lubricant of the sliding contacts. The CVFC model was addressed and developed for the modeling of the third body and to highlight both the simultaneous (deposition/exhaustion) lubrication process and the slider pad friction together with pellet wear on a tribometer with two simultaneous contacts (pellet/disk and slider pad/disk).

The lubricating film amount that covers the disk asperities is quantified (as the third body) by the fractional coverage and occurs in time.

The results suggest that asperities coverage plays a significant role in the interception of the tribological phenomena occurring on the separation surface lubricated with TiO_2_ powder, though it is not the only means of lubrication and velocity accommodation at the separation surface.

The CVFC model surprises qualitatively and quantitatively, in a way reasonable, the sliding contacts tribological behavior during the experiments and it is in concordance with experiments results of the pellet/disk in-line with the slider pad/disk.

At the same time, the model CVFC helps in an adequate way for friction coefficient evaluation at the slider pad/disk separation surface and the wear rate of the pellet.

Results of the CVFC model are used for analyzing the tribological behavior of sliding contacts lubricated with solid film and are compared with experimental results obtained on the pellet/disk in-line with slider pad/disk.

Theoretical and experimental results obtained have proved the excellent tribological capabilities of TiO_2_ powder with a normal evolution of the tribological parameters.

In effect, both the experiment and the CVFC model illustrate the self-repairing and self-replenishing attributes of a TiO_2_ powder lubrication scheme.

The closeness between the model and experiment is promising, in general.

## Figures and Tables

**Figure 1 materials-15-07145-f001:**
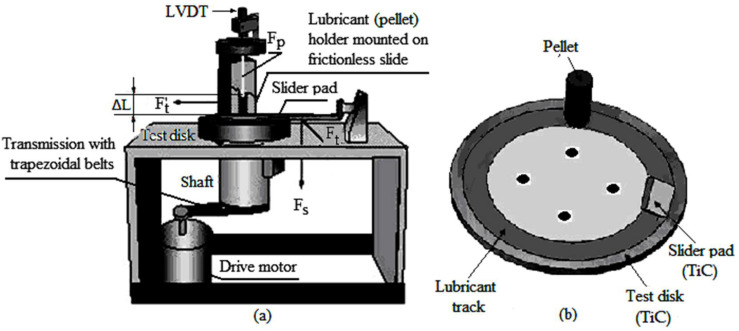
Experimental pellet/disk and pad/disk setup for evaluation of the friction forces pellet/disk at and slider pad/disk, and pellet wear (**a**) and schematic setup pellet/disk with slider pad/disk (**b**).

**Figure 2 materials-15-07145-f002:**
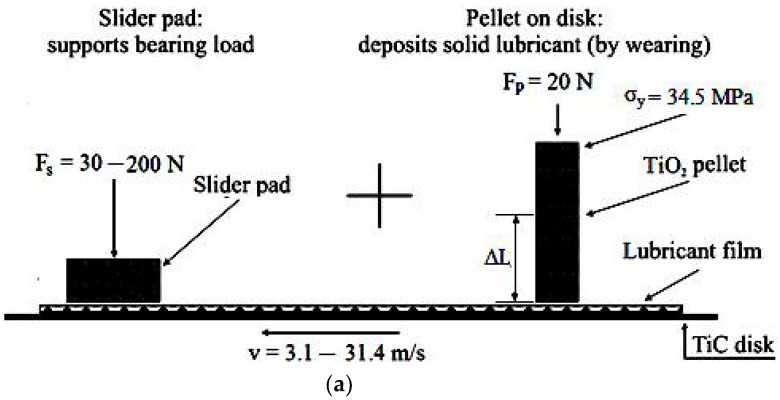
Schematic setup of the pellet/disk in line with a slider pad/disk for the transferred powder film study with self-replenishing and self-repairing (**a**) and of the pellet sheared intentionally against the TiC disk (**b**).

**Figure 3 materials-15-07145-f003:**
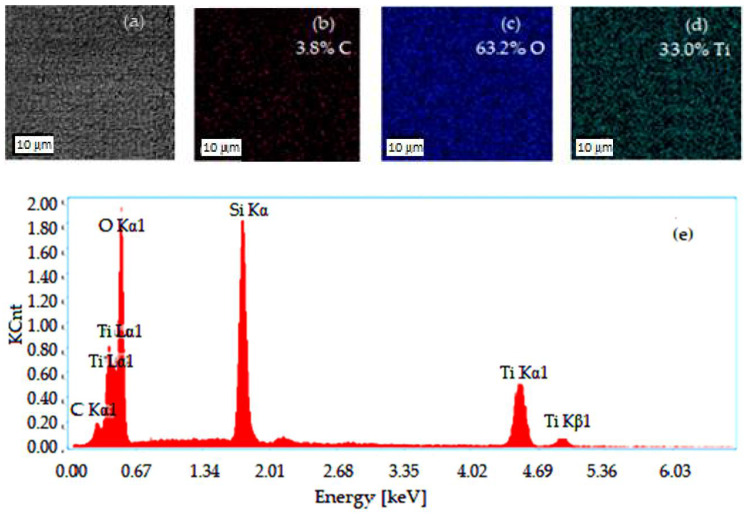
EDS analysis results of TiO_2_ powder film (**a**); the EDS mapping of C (**b**); the EDS mapping of O (**c**); the EDS mapping of Ti, as elements of the TiO_2_ film deposited on TiC disk (**d**); the EDS spectrum of the area shown in Figure 3a (**e**).

**Figure 4 materials-15-07145-f004:**
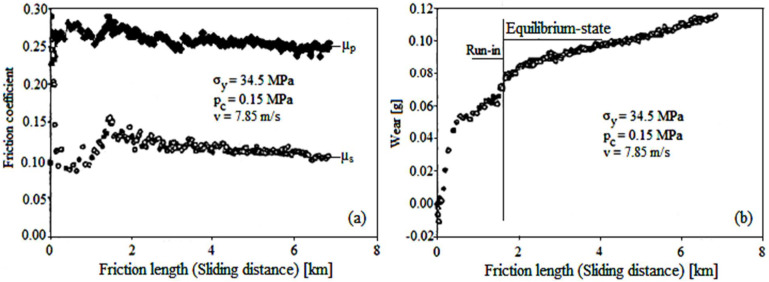
Experimental results on frictional and wear behavior of slider pad/disk in same conditions of tension, σ_y_, pressure, p_c_, and velocity, v: friction coefficient variation (**a**) and of wear intensity gravimetrical (**b**) depending on friction length (sliding distance). Noted: The friction coefficient, μ_p_, and μ_s_ were determined from the ratio, μ_p_ = F’_t_ /F_p_, respectively, the ratio, μ_s_ = F_t_/F_s_ (F’_t_ and F_t_—friction forces, measured at the pellet base and of slider pad base, respectively, F_p_ and F_s_—normal load of the pellet and of the slider pad (see Figure 1)).

**Figure 5 materials-15-07145-f005:**
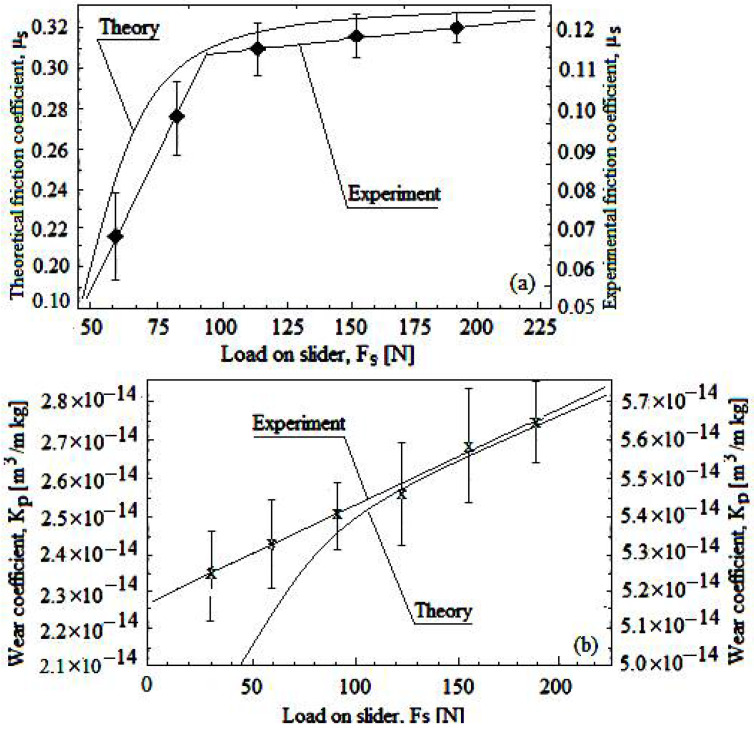
Variation of friction coefficient, μ_s_ (**a**), and of wear rate, K_p_ (**b**) depending on slider pad load at the separation surfaces (theoretical and experimental).

**Figure 6 materials-15-07145-f006:**
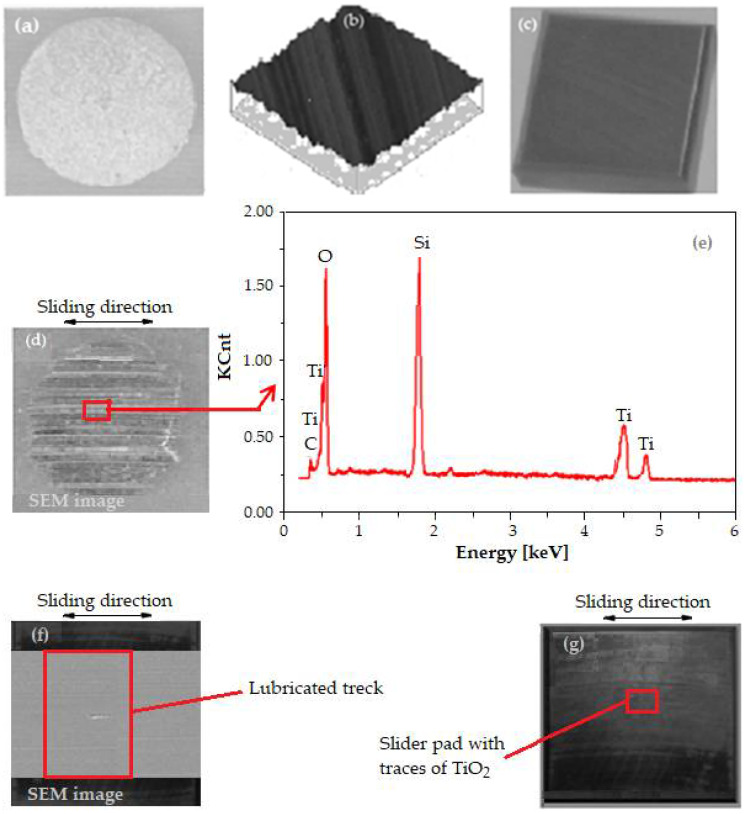
Images before testing (**a**–**c**) and after testing (**d**–**f**): (**a**) photography of the TiO_2_ pellet (front view), (**b**) AFM image from an area of the route (contact) where the lubrication track is made on the TiC disk, (**c**) photograph of the TiC slider pad; (**d**) SEM micrograph of the TiO_2_ pellet, (**e**) EDS pattern of area in position (**d**) from Figure 6; (**f**) SEM micrograph of the lubricated track; (**g**) photograph of the TiC slider pad.

## Data Availability

Data is contained within the article.

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
