# Peer review of "Tribological Properties Study of Solid Lubrication with TiO2 Powder Particles"

_materials, 2022, doi:10.3390/ma15207145_

Round 1

Reviewer 1 Report

I think the innovation of articles is insufficient. Although it has practical application in friction and wear, the readability does not meet the standard of publication. In addition, the expression of the article does not conform to the traditional expression method of the journal, and can be further improved. Besides, the figure quality in the article is rough, there is still a certain gap with the standards of publication. 

The mechanism part of the article can be simply explained, but cannot be missing, which can pull down the theoretical depth of the article.

I suggest that the author(s) re-submit the manuscript with major revisions.

Author Response

Response Letter

for Reviewer 1, Round 1

Manuscript ID materials-1930252

          entitled:

"Tribological Properties Study of Solid Lubrication with TiO2 Powder Particles"

                                                                            by

Filip Ilie, George Ipate, Florentina Cristina Manaila

Note: The manuscript is very colorful. What was deleted is marked in red and cut, and what was entered is marked in green.

Thanks to Reviewer 1 for the comments made, which are presented too generalized!  Exactly, I don't know what to add or delete!

However, after analyzing deeper the paper, I think realized what you meant!

As proof, you will notice a lot of changes and additions and I hope it’s okay now!

Thus:

  1. Reviewer 1 says: I think the innovation of articles is insufficient. Although it has practical application in friction and wear, the readability does not meet the standard of publication. In addition, the expression of the article does not conform to the traditional expression method of the journal, and can be further improved. Besides, the figure quality in the article is rough, there is still a certain gap with the standards of publication.

Response 1: In order to improve the content of the article and meet the standard of publication according to the traditional method of the journal, the authors have revised the manuscript to meet these requirements, as follows:

- first of all, chapter 2 'Materials, experiments, and research method' was divided into 'Materials and method' and 'Experimental procedure' with additions of new paragraphs and cancellation of others, and the other chapters renumbered and also with additions and cancellations of paragraphs or sentences, for that the manuscript to fulfill journal requirements (see  manuscript);

- secondly, two more figures were introduced with the inherent explanations and renumered all the figures. Then, there are many other paragraphs canceled and completed with new or added ones, etc. (see  manuscript);

- thirdly, the quality of the figures has been improved to meet publication standards (see manuscript).

  1. Reviewer 1 says: The mechanism part of the article can be simply explained, but cannot be missing, which can pull down the theoretical depth of the article.

Response 2: I hope that through everything that has been added and cancelled, the mechanism part of the article was explained and simplified, without affecting the theoretical depth of the manuscript.

So, please you to review the entire manuscript, because it has a new guise and presentation!

Thanks for your understanding!

Date: Sept. 26, 2022                                                                                                                                                                                                                  Authors

Reviewer 2 Report

The article ”Tribological Properties Study of Solid Lubrication with TiO2 Powder Particles” can be published in Materials Journal just after some minor corrections:

The authors should improve the followings:

- in the introduction, some specific values of mechanical properties should be presented by other researches

- please add indentation analysis and also SEM images of the friction tests. These results are not complete without some SEM images.

- add extra references

Author Response

Response Letter

for Reviewer 2, Round 1

Manuscript ID materials-1930252

          entitled:

"Tribological Properties Study of Solid Lubrication with TiO2 Powder Particles"

                                                                            by

Filip Ilie, George Ipate, Florentina Cristina Manaila

Note: The manuscript is very colorful. What was deleted is marked in red and cut, and what was entered is marked in green.

Many thanks to Reviewer 2 for the comments made. In addition, we made the indicated improvements, namely:

  1. Reviewer 2 says: - in the introduction, some specific values of mechanical properties should be presented by other researches.

Response 1: Thanks for the suggestion and as you will notice, several specific values of the mechanical properties (the most frequently encountered) presented by other researches (works) have been introduced in Introduction in a sentence „The specific values of mechanical properties ... oil dispersion(Hegman) ≥ 6.00; dry powder witheness ≥ 95 %”  (see the manuscript).

  1. Reviewer 2 says: - please add indentation analysis and also SEM images of the friction tests. These results are not complete without some SEM images.

Response 2: Thank you for the indication and you will notice that two new figures have been introduced (Figure 3 in chapter 3 "Experimental procedure" and Figure 6 in chapter 4 "Results and discussion") with images (EDS analysis, including SEM images) with the explanations and completions, as such (see manuscript).

  1. Reviewer 2 says: - add extra references

Response 3: You will notice that new references have been introduced, marked in green and as such, the numbering has changed.

Thanks for your understanding!

Date: Sept. 26, 2022                                                                                                                                                                                                                    Authors

Reviewer 3 Report

This work is useful for tribology scientist. However, the now version is bad to be understanded, especially such as the language and figures. I recommend the authors to carefully modify it. Some important questions should be clarified or improved.

1) the language, particularly grammar and logic, needs to be improved by a native english writing.

2) nealr all the figures are in the very low resolution, which is not readable.

3) the background about the tribology or lubrication of power/particle is inadequant, some latest references (up to now) are not cited in introduction section, and also there are not references published in the journal of materials. At least the following papers shoud be cited: (1) Comparative Study of Rheological Effects of Vegetable Oil-Lubricant, TiO2, MWCNTs Nano-Lubricants, and Machining Parameters' Influence on Cutting Force for.. (2) Synthesis and tribological properties of MXene/TiO2/MoS2 nanocomposite. (3) Performance analysis of journal bearing operating on nanolubricants with TiO2, CuO and Al2O3 nanoparticles as lubricant additives. (4)Investigation on the robust adsorption mechanism of alkyl-functional boric acid nanoparticles as high performance green lubricant additives. and so on. 

Author Response

Response Letter

for Reviewer 3, Round 1

Manuscript ID materials-1930252

          entitled:

"Tribological Properties Study of Solid Lubrication with TiO2 Powder Particles"

                                                                            by

Filip Ilie, George Ipate, Florentina Cristina Manaila

Note: The manuscript is very colorful. What was deleted is marked in red and cut, and what was entered is marked in green.

Many thanks to Reviewer 3 for the comments and recommendations made. In addition, we made the indicated improvements, namely:

  1. Reviewer 3 says: the language, particularly grammar and logic, needs to be improved by a native english writing.

Response 1: Thank you for the observation, but I inform you that the English language and grammar have been improved with the help of a native English speaker, as you will notice by the corrections made (see the manuscript)

  1. Reviewer 3 says: near all the figures are in the very low resolution, which is not readable.

Response 2: All the figures have been rechecked and the resolution has been increased. (see manuscript).

  1. Reviewer 3 says: - the background about the tribology or lubrication of power/particle is inadequant, some latest references (up to now) are not cited in introduction section, and also there are not references published in the journal of materials. At least the following papers shoud be cited: (1) Comparative Study of Rheological Effects of Vegetable Oil-Lubricant, TiO2, MWCNTs Nano-Lubricants, and Machining Parameters' Influence on Cutting Force for.. (2) Synthesis and tribological properties of MXene/TiO2/MoS2 nanocomposite. (3) Performance analysis of journal bearing operating on nanolubricants with TiO2, CuO and Al2O3 nanoparticles as lubricant additives. (4)Investigation on the robust adsorption mechanism of alkyl-functional boric acid nanoparticles as high performance green lubricant additives. and so on. 

Response 3: You will notice that new references have been introduced more new references, first of all the ones indicated by you (for which I thank you) and then with references published in the materials journal, marked in green, and as such, the numbering has changed.

Thanks for your understanding!

Date: Sept 26, 2022                                                                                                                                                                                                                   Authors

Reviewer 4 Report

The manuscript is well written and the authors have nicely presented the impressive amount of data derived from the experiments. I recommend "accept after minor revision".

 Comments:

1) What is the advantage of solid lubricants (powder lubricants) with respect to water-based lubricants? Please explain that in the introduction.

2) The authors should compare the Tribological Properties of TiO2 powder with other solid powders such as MoS2, and WS2 reported in literature.

Author Response

Response Letter

for Reviewer 4, Round 1

Manuscript ID materials-1930252

          entitled:

"Tribological Properties Study of Solid Lubrication with TiO2 Powder Particles"

                                                                            by

Filip Ilie, George Ipate, Florentina Cristina Manaila

Note: The manuscript is very colorful. What was deleted is marked in red and cut, and what was entered is marked in green.

Many thanks to Reviewer 4 for appreciation and the comments made. In addition, we made the indicated improvements, namely:

  1. Reviewer 4 says: What is the advantage of solid lubricants (powder lubricants) with respect to water-based lubricants? Please explain that in the introduction.

Response 1: Thank you for the suggestion, which I achieved by introducing a paragraph with the required explanations, namely: "Solid lubricants (powder lubricants) have some advantages over water-based lubricants, ... since the absorption of water molecules between the layers of solid lubricants further reduce the forces of attraction between the layers." (see the manuscript).

  1. Reviewer 4 says: The authors should compare the tribological Properties of TiO2 powder with other solid powders such as MoS2, and WS2 reported in literature.

Response 2: This comparison was made by presenting values of the friction coefficient as one of the most important tribological parameters, by adding the paragraph (marked in green) " In order to have a comparative picture from the point of view of tribological properties ... especially in extreme conditions.", also in chapter 1 'Introduction' (see manuscript).

Thanks for your understanding!

Date: Sept. 26, 2022                                                                                                                                                                                                                       Authors

Round 2

Reviewer 1 Report

The quality of the article has been improved to a certain extent after the revision, but some content still needs minor revision.

1. The expression of wear coefficient in the conclusion section is not rigorous, it is generally called wear rate. This expression should be unified throughout the article.

2.  Figure 6 does not clearly show the traces and evidence of self-repair performance, it is recommended to replace and add clearer figures.

3. The mechanism section in the article is not enough,which can be appropriately added and displayed in the form of mechanism diagram. For details, refer to the article Nanomaterials 2020, 10, 200; doi:10.3390/nano10020200.

After the above modifications, I suggest that it can be accepted and published.

Author Response

Response Letter

for Reviewer 1, Round 2

Manuscript ID materials-1930252

          entitled:

"Tribological Properties Study of Solid Lubrication with TiO2 Powder Particles"

                                                                            by

Filip Ilie, George Ipate, Florentina Cristina Manaila

Note: The manuscript is very colorful. What was deleted is marked in red and cut, what was entered in Round 1 is marked in green, and what was introduced in Round 2 is marked in blue.

Thanks to Reviewer 1 for the new comments made, but to your observation from point 3, I didn't understand exactly what you mean, when you say, "The mechanism section in the article ...", which mechanism section? I think that, it was simple to specify the section number or the name of the section in the article!

However, I completed Figure 6 with a diagram (current position (d)) and other additions (see Figure 6), along with the additions to your requirements from point 1 and 2 (see manuscript).

Regarding point 2, you will notice that the current position (f) in Figure 6 that has been changed and I hope that the traces and evidence of self-repair performance to be seen much more clearly (see manuscript).

In the hope that my answers will be in accordance with your observations and requirements!

Thus:

  1. Reviewer 1 says: The expression of wear coefficient in the conclusion section is not rigorous, it is generally called wear rate. This expression should be unified throughout the article.

Response 1: Thank you for your observation and I inform you that I have changed the entire article (marked in blue, see the manuscript).

  1. Reviewer 1 says: Figure 6 does not clearly show the traces and evidence of self-repair performance, it is recommended to replace and add clearer figures.

Response 2: Thank you for the observation and by adding an additional position and the changes made to positions d, f and g, respectively by the additions added to the text, I hope to have answered appropriately (see the manuscript).

  1. Reviewer 1 says: The mechanism section in the article is not enough,which can be appropriately added and displayed in the form of mechanism diagram. For details, refer to the article Nanomaterials 2020, 10, 200; doi:10.3390/nano10020200.

Response 3: Through the above presented and the additions made to the "Results and Discussions" section, respectively consulting the article 'Nanomaterials 2020, 10, 200; doi:10.3390/nano10020200' I hope we managed to agree with you as much as possible (see the manuscript).

Thanks for your understanding!

Date: Oct. 02, 2022                                                                                                                                                                                                                        Authors

Reviewer 2 Report

Now the paper is suitable for publication.

Reviewer 3 Report

All figuress in my review version is not clear, please ensure they are in high resolution. I have no other questions.

Author Response

Response Letter

for Reviewer 3, Round 1

Manuscript ID materials-1930252

          entitled:

"Tribological Properties Study of Solid Lubrication with TiO2 Powder Particles"

                                                                            by

Filip Ilie, George Ipate, Florentina Cristina Manaila

Note: The manuscript is very colorful. What was deleted is marked in red and cut, and what was entered is marked in green, after Round 1 and with blue, after Round 2.

Many thanks, to Reviewer 3, for the recommendation. In addition, I enlarged resolution the figures and I hope it’s okay now (see manuscript)!

  1. Reviewer 3 says: All figuress in my review version is not clear, please ensure they are in high resolution. I have no other questions.

Response 1: All the figures have been rechecked and the resolution has been increased, some even with additions (see manuscript).

Thanks for your understanding!

Date: Oct 02, 2022                                                                                                                                                                                                                       Authors

Round 3

Reviewer 1 Report

This article has undergone a lot of modifications. Although it has some content and innovation to be improved, it can be published.